# Efficacy of Four In Vitro Fungicides for Control of Wilting of Strawberry Crops in Puebla-Mexico

Alba Cruz Coronel [1], Conrado Parraguirre Lezama [1], Yesenia Pacheco Hernández [1], Olga Santiago Trinidad [2], Antonio Rivera Tapia [3] and Omar Romero-Arenas [1,*]

1   Centro de Agroecología, Instituto de Ciencias, Benemeérita Universidad Autoónoma de Puebla, Edificio VAL 1, Km 1, 7 Carretera a San Baltazar Tetela, San Pedro Zacachimalpa, Puebla 72960, Mexico; alba.cruzcor@alumno.buap.mx (A.C.C.); conrado.parraguirre@correo.buap.mx (C.P.L.); yesenia.pachecoh@gmail.com (Y.P.H.)
2   Campo Experimental El Palmar, CIR-Golfo Centro, Instituto Nacional de Invistigaciones Forestales Agrícolas y Pecuarias (INIFAP), Tezonapa, Veracruz 95083, Mexico; santiago.olga@inifap.gob.mx
3   Centro de Investigaciones en Ciencias Microbiológicas, Instituto de Ciencias, Benemérita Universidad Autónoma de Puebla, Ciudad Universitaria, Puebla 72570, Mexico; jose.riverat@correo.buap.mx
*   Correspondence: biol.ora@hotmail.com; Tel.: +52-22-2229-5500 (ext. 3717)

**Abstract:** Strawberry wilt is an established disease of strawberry crops caused by fungus *Fusarium solani*. In Mexico, strawberry cultivation represents an important productive activity for several rural areas; however, wilt disease affects producers economically. The objectives of this research were: (a) to identify and morphologically characterize strain "MA-FC120" associated with root rot and wilting of strawberry crops in Santa Cruz Analco, municipality of San Salvador el Verde, Puebla-Mexico; (b) to evaluate the potential of single and multiple applications of four broad-spectrum fungicides used against *F. solani* in vitro. Plant tissue samples were collected from strawberry crops in Puebla-Mexico with presence of symptoms of desiccation and root rot. Strain "MA-FC120" was identified as *F. solani*, being the causal agent of wilt and root rot in strawberry plants from Santa Cruz Analco. Fungicide Benomyl 50® showed the highest percentage of inhibition on *F. solani* (100%) under in vitro conditions. The fungicide Mancosol 80® and Talonil 75® at low concentration (600 and 450 mg L$^{-1}$) showed no toxicity, being harmless to strain MA-FC120. However, fungicide Talonil 75® showed slight toxicity at the dose recommended by the manufacturer and moderate toxicity in high concentration (1350 mg L$^{-1}$). Likewise, Captan 50® in its three concentrations evaluated showed slight toxicity, obtaining around 50% on the classification scale established by International Organization for Biological Control (IOBC).

**Keywords:** fungicide resistance; PCR; broad-spectrum fungicides; *Fusarium solani*; toxicity

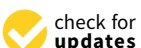



## 1. Introduction

Strawberry (*Fragaria × ananassa*) is a widely distributed crop in the world due to its genotypic diversity, highly heterozygous nature, and wide range of environmental adaptations [1,2]. Strawberry is a favored food due to its nutraceutical properties and antioxidant effects. It is also a relevant source of bioactive compounds, due to its high level of vitamin C and phenolic compounds that provide beneficial effects on maintenance of consumers' health, especially by strengthening the immune system [3].

World strawberry production in 2021 was 12,106,585 tons, with China being the main producer with 3,213,000 tons, followed by USA (1,164,000 tons) and Mexico [4,5]. The cultivated area in Mexico is currently 12,913 ha, with a production of 557,514 tons in 2020 [6]. The State of Puebla ranks ninth in national production [7].

The strawberry crop has several fungal, bacterial, and viral diseases which cause large yield losses, with fungal diseases being of greatest concern and causing huge eco-nomic damage [8,9]. Soil-borne fungal pathogens such as *Fusarium oxysporum*, *F. solani*,

*Macrophomina phaseolina*, *Pythium* spp., *Phytophthora* spp., and *Rhizoctonia* spp., cause the death of feeder rootlets, plant deterioration, and blackening of the main root system, as well as a decrease in plant stand vigor and productivity, causing a decrease in crop yield [10,11].

Among the fungal diseases of the strawberry crop, wilting or drying is a disease caused by *F. solani* (Mart) Sacc, causing root rot in the crop, which causes stunting, wilting, and death of plants [12,13]. *F. solani* is a generalist plant pathogenic fungus with soil and seed origin present worldwide [14]. It is mainly characterized by abundant aerial mycelium, cottony texture, and sickle-shaped macroconidia with two to five septa [15]. *F. solani* has recently been reported in Spain in strawberry crops in intensive "nursery" systems, as well as in production fields [16]. *F. solani* has also been described as a pathogen of strawberry in Italy [17], Iran [18], and Pakistan [19]. However, there are few reports in Mexico [8].

Multiple fungicides to control various fungal diseases in strawberry crops, including demethylation inhibitors, quinone external inhibitors, and succinate dehydrogenase inhibitors (Fungicide Resistance Action Committee codes 3, 11, and 7, respectively), are available to producers. Benomyl and thiophanate-methyl have been consistently reported to be effective against various species of the genus *Fusarium* on several host plants [20,21]. Chemical management is an important tool for control of diseases, including soil-borne diseases. In addition, identification of effective fungicides would allow consolidation of different components needed to formulate integrated fungicides for disease management [22].

In Mexico, there is no publicly available information on the name, type, and quantity of each authorized pesticide applied nationally [23]. According to García et al. [24], the active ingredients (a.i.) Mancozeb®, Benomyl®, Cholothalonil®, and Captan® can be identified as most widely used in the country. Given this situation, chemical control is considered an efficient practice; however, extensive and repeated application of these chemicals has led to the frequent appearance of resistance to fungicides that could compromise their effectivity, as highlighted by incidences of resistance to carbendazim and tebuconazole [25].

In the summer of 2021, plant stunting, wilting, leaf chlorosis, and root rot were observed on strawberry var. "Camino Real" in Santa Cruz Analco, Puebla-Mexico, with symptoms indicative of *F. solani*. These symptoms were observed in 40% of the crops. Therefore, the objectives of this research were: (a) to identify the causal agent of this disease; (b) to evaluate the potential of single and multiple applications of four broad-spectrum fungicides used against *F. solani* in vitro.

## 2. Materials and Methods

### 2.1. Isolation Area

Samples of plant tissue with stem and root rot were collected from a 1000 m$^2$ plot of strawberry var. "Camino Real" during summer–autumn 2021 production. The agricultural plot corresponds to the community of Analco de Ponciano Arriaga (Santa Cruz Analco), which belongs to the municipality of San Salvador el Verde in the State of Puebla, Mexico. The study community has a temperate sub-humid climate (Cw) and average rainfall of 1000 mm [26]. Sampling was directed towards individuals with symptoms associated with genus *Fusarium*; all samples were kept in plastic bags in a cooler at 4 °C until they were transferred to the laboratory.

### 2.2. Isolation of F. solani from Plant Tissues

Samples were cut into small 5 mm pieces, surface-sterilized with 1% sodium hypochlorite for 3 min, and rinsed in sterile distilled water three times. Pieces were placed on potato dextrose agar medium (PDA, Bioxon, Becton Dickinson and Company, Queretaro, Mexico) and incubated at 28 °C for 10 days. The developed colonies were isolated and purified by monosporic cultures, which involved the transfer of a single conidium to potato dextrose agar (PDA, Bioxon, Becton Dickinson and Company, Queretaro, Mexico) medium. Subsequently, the plates were transferred to a microbiological incubator at 24 °C (Thermo Scientific, CA, USA) and subjected to a photoperiod of 12:12 L:D (12 h of light: 12 h of darkness) for 10 days [2].

### 2.3. Morphological Characterization

Identification of fungal colonies obtained was performed by analyzing morphological characteristics associated with genus *Fusarium*, using taxonomic identification keys from Barnett and Hunter [27], in a microculture system in an optical microscope (Carl Zeiss, Jena, Germany) at $1000\times$ magnifications.

Once the isolates were characterized, the most representative was selected based on its ability to grow in three growth media tested: (a) potato dextrose agar medium (PDA, Bioxon, Becton Dickinson and Company, Queretaro, Mexico), (b) tryptone soybean agar (TSA, Sigma-Aldrich, Mexico [28]), and (c) water agar with 8 g/L carnation (CWA [29]). All culture media were incubated with a photoperiod of 12:12 L:D (12 h of light: 12 h of darkness) for 10 days [2]. Likewise, the diameter of the mycelium was measured every 12 h with a digital vernier caliper (CD-6 Mitutoyo) to estimate the growth rate (mm d$^{-1}$) [30].

### 2.4. DNA Extraction, PCR Amplification, and Sequencing

DNA was extracted from conidia, conidiophores, and mycelium of isolates obtained with morphological characteristics associated with *F. solani*. This procedure was performed with 2% cetyl trimethylammonium bromide (CTAB) [31]. Genomic DNA was suspended in 100 µL of sterile Milli-Q water and quantified by spectrophotometry in a Nano Drop 2000c (Thermo Scientific, Waltham, MA, USA). To determine the DNA quality, absorbance values between 1.8 and 2.2 at A$_{80}$/$_{260}$ and A$_{230}$/$_{260}$ nm were considered acceptable. Finally, the DNA was diluted to 20 ng µL$^{-1}$ and then stored at $-20\,^{\circ}$C for until further processing.

From the most representative colonies present in diseased strawberry plants, isolate 20 from plot 1 was chosen and named strain MA-FC120. Molecular identification of strain MA-FC120 was carried out based on the analysis of internal transcribed spacer (ITS) region sequences using primer pairs ITS5/ITS4 [32], and by sequencing a part of translation elongation factor-1 alpha (EF-1α) using primers EF-1 and EF-2 [16]. Amplified PCR products were verified by electrophoresis on a 1.5% agarose gel (Seakem, CA, USA). They were then purified and ligated into pGEM T-Easy Vector (Promega) and bidirectionally sequenced by Macrogen, Seoul, Korea. Sequences were assembled and edited using SeqMan (DNAStar, Madison, WI, USA) and compared to established sequences in GenBank™ using the Blast algorithm.

Phylogenetic analyses were performed with concatenated sequence alignment of genes ITS/TEF-1α using MEGA X software [33]. The analysis involved 11 nucleotide sequences obtained from the GenBank™ database (Table 1). Sequences from the ITS and TEF-1α regions of *Fusarium oxysporum* were used as an outgroup. Evolutionary history was inferred using the maximum likelihood method and the Tamura–Nei model [34], applying the Neighbor-Join and BioNJ algorithms to a distance matrix using the all-sites option. There was a total of 17,513 pb in the final dataset per 1000 bootstrap replicates to assess relative branch stability.

### 2.5. Pathogenicity Tests

Fifty Camino Real strawberry plants provided by local farmers were used. Each two-month-old plant was planted individually in a 1 L plastic pot containing a sterilized mixture of Peatmoss and Agrellite (1:1 *v/v*). Inoculation of strain MA-FC120 was performed by direct spray to runoff with $1\times10^6$ conidia/mL conidia suspension from pure cultures, where they had already developed macroconidia; these were taken using a 0.5–1 µL micropipette in a laminar flow hood, added to 10 mL of sterile physiological saline solution and incubated for 7 days at 25 °C [30]. Plant development was carried out under greenhouse conditions (25 °C and 70% relative humidity) until the appearance of disease symptoms. For the case of the control group, 50 healthy plants were used that were inoculated by direct spray to runoff with sterile water and kept under the same greenhouse conditions.

**Table 1.** *Fusarium solani* species complex gene (FSSC) bank sequences to corroborate the identity of strain MA-FC120.

| Species (FSSC) | Strain | Isolation Source | Country | No. Access | |
|---|---|---|---|---|---|
| | | | | ITS | EF-1α |
| *Fusarium solani* | MA-FC120 | Strawberry | México | OM473287 | OM616884 |
| *Fusarium solani* | SVY-402-1 | Pea | EU | KJ437436 | KM044428 |
| *Fusarium solani* | GuangX17 | Bitter grand | China | KY785014 | KY785024 |
| *Fusarium solani* | CB24-4 | Ginseng | China | MN637861 | MN650097 |
| *Fusarium solani* | MJ | Strawberry | Spain | MH300474 | MH300509 |
| *Fusarium solani* | MY | Strawberry | Spain | MH300482 | MH300517 |
| *Fusarium solani* | MS | Strawberry | Spain | MH300452 | MH300523 |
| *Fusarium solani* | MRC 2635 | Wheat | India | MH582403 | MH582423 |
| *Fusarium solani* | MRC 2805 | Wheat | India | MH582404 | MH582424 |
| *Fusarium solani* | NRRL 52778 | Chinch | Syria | JF740931 | JF740846 |
| *Fusarium oxysporum* | Esm3 | Golden berry | Colombia | KJ936621 | JX465113 |

## 2.6. In Vitro Sensitivity Bioassay

The efficiency of a systemic fungicide Benomyl 50® and three protectors, Talonil 75®, Mancosol 80®, and Captan 50® with three concentrations per product was evaluated: (a) commercial dose authorized by the manufacturer; (b) half the recommended dose, and; (c) double the commercial dose expressed in mg L$^{-1}$ (Table 2).

**Table 2.** Fungicides at different concentrations evaluated.

| Fungicide (Commercial Name) | Active Ingredient | Molecular Formula | Concentration (mg L$^{-1}$) | | |
|---|---|---|---|---|---|
| | | | Low | Recommended | High |
| Control | Water | $H_2O$ | - | - | - |
| Captan 50® | Captan | $C_9H_8Cl_3NO_2S$ | 450 | 900 | 1350 |
| Mancosol 80® | Mancozeb | $C_4H_6MnN_2S_4$ | 600 | 1200 | 1800 |
| Talonil 75® | Chlorothalonil | $C_8Cl_4N_2$ | 450 | 900 | 1350 |
| Benomyl 50® | Benomyl | $C_{14}H_{18}N_4O_3$ | 250 | 500 | 750 |

## 2.7. Mycelial-Growth Inhibition

The controlled-poisoning technique [35] was used, which consisted of placing 5 mm-diameter discs with mycelium of pathogens in the center of the Petri dish with potato dextrose agar medium (PDA, Bioxon, Becton Dickinson and Company, Queretaro, Mexico) plus fungicide at different concentrations (Table 2), which were incubated in dark conditions at 28 °C for 10 days. Mycelial diameter was measured every 12 h with a digital vernier (CD-6 Mitutoyo) to estimate growth rate (mm d$^{-1}$), which was calculated with the linear growth function [30] Equation (1), obtaining the average of four measurements of the longitudinal diameter per experimental unit.

$$Y = MX + B \tag{1}$$

where:

Y = distance
M = pending
X = time
B = the constant factor.

As a control, pathogen discs were used on PDA medium without fungicides. Evaluation was concluded when the mycelium of the pathogen completely covered the plate of control treatment. Each treatment consisted of three replicates and was carried out

in duplicate in a completely randomized experimental design [36]. The mycelial-growth inhibition percentage (I%) was calculated with the Abbot formula [37] (2):

$$(I\%) = (C-T)/C \times 100 \tag{2}$$

where:

I% = percent growth reduction in *F. solani*
C = colony growth (cm) in the control
T = colony growth (cm) in the treatment.

To determine the toxicity according to inhibition percentages obtained, the results were classified based on the classification scale established by the IOBC [38] (Table 3).

**Table 3.** Toxicity classification scale [38] (Reproduced with permission from Viñuela et al., Phytoma; published by PHYTOMA-España, 1993).

| Growth Inhibition (%) | Classification |
| --- | --- |
| <30 | Harmless |
| 30–75 | Slightly toxic |
| 75–90 | Moderately toxic |
| >90 | Toxic |

*2.8. Statistical Analysis*

Statistical analyses were performed with IBM SPSS Statistics version 25 software, using a completely randomized factorial design. The variable (I%) was expressed as a percentage and transformed with angular arccosine $\sqrt{x} + 1$. Data were analyzed with analysis of variance (MULTIVARIANT ANOVA) using a quadratic model of response as a function of mycelial-growth inhibition percentage (I%), development rate (mm h$^{-1}$), and growth rate (mm d$^{-1}$) to determine significant differences among treatments, under the following mathematical model: (3):

$$I\gamma j = \mu + ti + \varepsilon ij \tag{3}$$

where:

I$\gamma$j = value of response variable of experimental unit associated with the $\gamma$-th treatment, and to the j-th repetition
$\mu$ = corresponds to general average of response variable in the experiment
ti = effect of $\gamma$-th treatment
$\varepsilon\gamma$j = error of experimental unit associated with the $\gamma$-th treatment
j-th = repetition
$\gamma$ = 1, 2, 3, 4 . . . 13
j = 1, 2, 3
I = variable under study (I%).

## 3. Results

*3.1. Isolation, Characterization, and Identification of Causal Agent of Wilting on Strawberry Crops*

Ten representative isolates from 50 different plants developed pinkish-white colonies with granular and powdery textures after 10 days of incubation; no aerial mycelium was observed. Soma consisted of septate hyphae 2.579–9.58 (5.789) μm wide (*n* = 50), and some false heads 18.497–72.566 (29.906) μm in diameter were visualized. Kidney-shaped oval microconidia 9.178–14.791 (12.78) μm long × 3.649–4.473 (4.232) μm wide were observed (*n* = 50). Spindle-shaped, slightly curved macroconidia 15.054–33.729 (21.33) μm long × 4.482–7.336 (5.256) μm wide (*n* = 50) were also observed (Figure 1A–F).

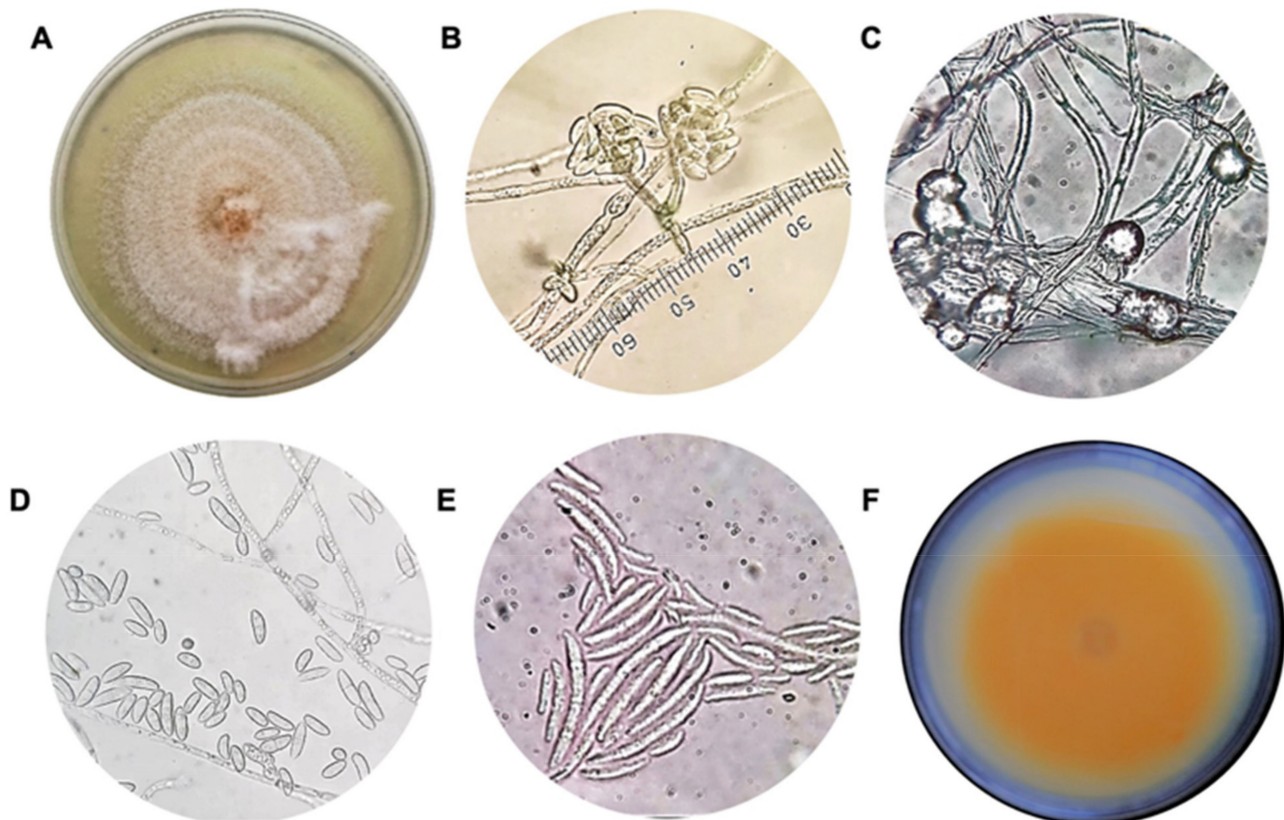

**Figure 1.** Morphology of the MA-FC120 strain: (**A**) Appearance of colony on PDA culture medium; (**B**) False heads of microconidia held in monophialides seen at 40× and 1000×; (**C**) Chlamydospora seen at 1000×; (**D**,**E**) microconidia and macroconidia seen at 400× and 100×, respectively; (**F**) Colony reverse staining on PDA.

Of the 10 characteristic isolates of *F. solani* from diseased strawberry plants, isolate MA-FC120 was chosen based on its high rate of development and speed of growth in three culture media tested (PDA, TSA, and CWA). The results obtained from the alignment of ITS gene sequences (551 bp) presented 100% identity with *F. solani*, as well as the elongation factor EF-1$\alpha$ (686 bp) confirmed the identity with *F. solani*, obtaining between 100 and 99, 85% identity, from data obtained from the nucleotide base of the GenBank™ of the National Center for Biotechnology Information.

The phylogenetic analysis generated from concatenated ITS/EF-1$\alpha$ sequences (Figure 2) clearly distinguishes at the species level, showing that strain MA-FC120 corresponds to the *Fusarium solani* species complex (FSSC).

### 3.2. Pathogenicity Tests

Koch's postulates confirmed that the MA-FC120 strain belonging to the *F. solani* species complex (FSSC) produced typical symptoms of desiccation, wilting, and root rot 20 days after inoculation. In addition, ascending wilt (Figure 3C), loss of turgor, and generalized vascular wilt (Figure 3B) were observed, finally leading to death of the entire plant 45 days after inoculation. No disease symptoms were observed in the control group (Figure 3A).

### 3.3. In Vitro Sensitivity Bioassay and Mycelial-Growth Inhibition

The treatments under study showed highly significant differences ($p = 0.0001$) in colony diameter, growth rate, and percentage inhibition, as well as interaction of the active ingredient of the different fungicide concentrations, at a 95% confidence level (Table 4).

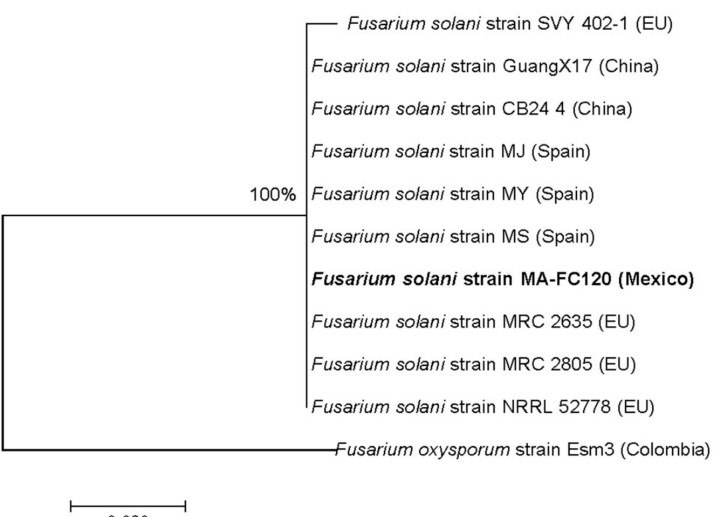

*Fusarium solani* strain SVY 402-1 (EU)

*Fusarium solani* strain GuangX17 (China)

*Fusarium solani* strain CB24 4 (China)

*Fusarium solani* strain MJ (Spain)

100% | *Fusarium solani* strain MY (Spain)

*Fusarium solani* strain MS (Spain)

**Fusarium solani** strain **MA-FC120** (Mexico)

*Fusarium solani* strain MRC 2635 (EU)

*Fusarium solani* strain MRC 2805 (EU)

*Fusarium solani* strain NRRL 52778 (EU)

*Fusarium oxysporum* strain Esm3 (Colombia)

0.020

**Figure 2.** The optimal tree with the sum of branch length = 0.1818 is shown. The tree is drawn to scale, with branch lengths in the same units as those of the evolutionary distances used to infer the phylogenetic tree generated from concatenated ITS/EF-1α sequences. *Fusarium oxysporum* was used as the outgroup. The MA-FC120 strains obtained in this study are shown in bold black.

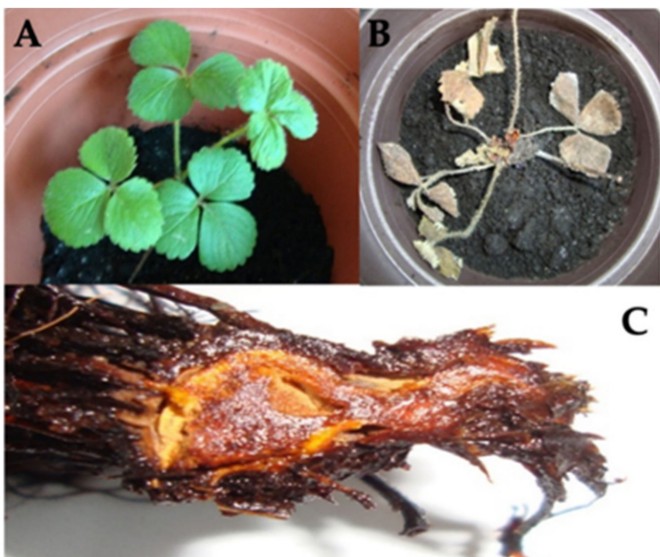

**Figure 3.** Pathogenicity tests on strawberry plants (var. Camino Real): (**A**) healthy control plant without symptoms of drying, (**B**) plant with wilting and death of foliage, (**C**) cross-section of strawberry root showing ascending root rot.

Table 5 shows the summary of the MULTIVARIATE ANOVA analysis for the quadratic response surface model, finding highly significant statistical differences. Model value for F = 523.500 (Treatments), 0.310 (Rate of Development), and 330.807 (Growth Rate), which implies a highly significant model for the percentage inhibition of mycelial growth (I%) in vitro.

All fungicides inhibited the growth rate of strain MA-FC120 compared to the control. However, systemic fungicide Benomyl 50® was the most effective product, presenting highly significant statistical differences with the protectant fungicides, inhibiting 100% of fungal development from the first days of incubation in the three doses applied (Figure 3).

The growth rate in the negative control group was 6.7104 mm d$^{-1}$. It was observed that, with the different concentrations of fungicides, it decreased considerably; however,

the systemic fungicide Benomyl 50® did not show growth after 10 days of incubation in the three doses applied (Table 4). The protective fungicide Talonil 75® at double the commercial dose (1350 mg L$^{-1}$) showed an inhibition percentage of 75.70% (Table 4) on mycelial growth and development. However, the protective fungicides Captan 50® and Mancosol 80® showed an inhibition percentage of 50% on the mycelial growth and development of strain MA-FC120 under in vitro conditions after ten days of incubation. It is worth mentioning that the fungicide Mancosol 80® at the lowest dose of 600 mg L$^{-1}$ showed a harmless effect on mycelial growth and development, obtaining 10.73%. Inhibition percentage (Table 4). In general, as the dose of the fungicides increased, an increase in the percentage of inhibition was observed.

**Table 4.** In vitro efficacy of four fungicides on the rate of development, speed of growth, and percentage of inhibition of mycelial growth of the MA-FC120 strain.

| Fungicides (Commercial name) | Concentration (mg L$^{-1}$) | Development Rate * (mm h$^{-1}$) | Growth Rate * (mm d$^{-1}$) | I% |
|---|---|---|---|---|
| Control | - | 0.2873 ± 0.0082 [d] | 6.7104 ± 0.0888 [g] | 0.00 [h] |
| Captan 50® | 450 | 0.1557 ± 0.0008 [bc] | 3.5296 ± 0.0833 [d] | 47.54 ± 3.12167 [e] |
| | 900 | 0.1426 ± 0.0077 [bc] | 3.2432 ± 0.1408 [d] | 51.71 ± 1.31683 [d] |
| | 1350 | 0.1366 ± 0.0030 [bc] | 3.1600 ± 0.0806 [d] | 52.90 ± 1.57778 [d] |
| Mancosol 80® | 600 | 0.2632 ± 0.0076 [d] | 5.9992 ± 0.0847 [f] | 10.73 ± 4.3486 [g] |
| | 1200 | 0.1658 ± 0.0114 [c] | 4.4512 ± 0.3004 [e] | 33.68 ± 5.1496 [f] |
| | 1800 | 0.1648 ± 0.0011 [c] | 4.3376 ± 0.0237 [e] | 35.46 ± 1.64257 [f] |
| Talonil 75® | 450 | 0.1427 ± 0.0279 [bc] | 4.7025 ± 0.7000 [e] | 29.95 ± 7.43654 [f] |
| | 900 | 0.0961 ± 0.0223 [b] | 2.4399 ± 0.5936 [c] | 63.48 ± 6.91699 [c] |
| | 1350 | 0.0923 ± 0.0244 [b] | 1.6305 ± 0.5031 [b] | 75.70 ± 9.65249 [b] |
| Benomyl 50® | 250 | 0.00 ± 0.00 [a] | 0.00 ± 0.00 [a] | 100.00 [a] |
| | 500 | 0.00 ± 0.00 [a] | 0.00 ± 0.00 [a] | 100.00 [a] |
| | 750 | 0.00 ± 0.00 [a] | 0.00 ± 0.00 [a] | 100.00 [a] |

* Means followed by the same letter are not significantly different for $p \leq 0.05$ according to Tukey test.

**Table 5.** Quadratic model of response surface for different concentrations of fungicides on rate of development, speed of growth, and percentage of inhibition of mycelial growth (I%) of the MA-FC120 strain in vitro.

| Origin | | Sum of Squares (Type III) | gL | Mean Square | F | Sig. |
|---|---|---|---|---|---|---|
| Corrected model | $X_1$ = Treatments | 523.500 [a] | 26 | 20.135 | 10.738 | <0.001 |
| | $X_2$ = Rate of Development | 0.310 [b] | 26 | 0.012 | 202.898 | <0.001 |
| | $X_3$ = Growth Rate | 330.807 [c] | 26 | 12.723 | 133.673 | <0.001 |
| Intersection | $X_1$ | 782.759 | 1 | 782.759 | 417.471 | <0.001 |
| | $X_2$ | 0.621 | 1 | 0.621 | 10580.715 | <0.001 |
| | $X_3$ | 477.343 | 1 | 477.343 | 5015.036 | <0.001 |
| I (%) | $X_1$ | 523.500 | 26 | 20.135 | 10.738 | <0.001 |
| | $X_2$ | 0.310 | 26 | 0.012 | 202.898 | <0.001 |
| | $X_3$ | 330.807 | 26 | 12.723 | 133.673 | <0.001 |
| Error | $X_1$ | 22.500 | 12 | 1.875 | | |
| | $X_2$ | 0.001 | 12 | $5.874 \times 10^{-5}$ | | |
| | $X_3$ | 1.142 | 12 | 0.095 | | |
| Total | $X_1$ | 1950.000 | 39 | | | |
| | $X_2$ | 0.937 | 39 | | | |
| | $X_3$ | 782.328 | 39 | | | |
| Total corrected | $X_1$ | 546.000 | 38 | | | |
| | $X_2$ | 0.311 | 38 | | | |
| | $X_3$ | 331.949 | 38 | | | |

(a) R-squared = 0.959 (R-squared adjusted = 0.870); (b) R-squared = 0.998 (R-squared adjusted = 0.993); (c) R-squared = 0.997 (R-squared adjusted = 0.989).

The effect of fungicides on mycelial development and morphology of fungal colonies of strain MA-FC120 was corroborated after 10 days of incubation with respect to control, showing a marked inhibition of mycelial development and pronounced radial growth deformation compared to the control group (Figure 3). However, strain MA-FC120 showed higher resistance to Mancosol 80® and Talonil 75® at lower doses (Figure 3). At the 600 mg $L^{-1}$ concentration, the strain reached a 5.9992 mm $d^{-1}$ development rate, while at the 450 mg $L^{-1}$ concentration, strain MA-FC120 reached a 4.7025 mm $d^{-1}$ development rate, in both cases presenting a harmless effect on the classification scale established by IOBC [38] (Figure 4).

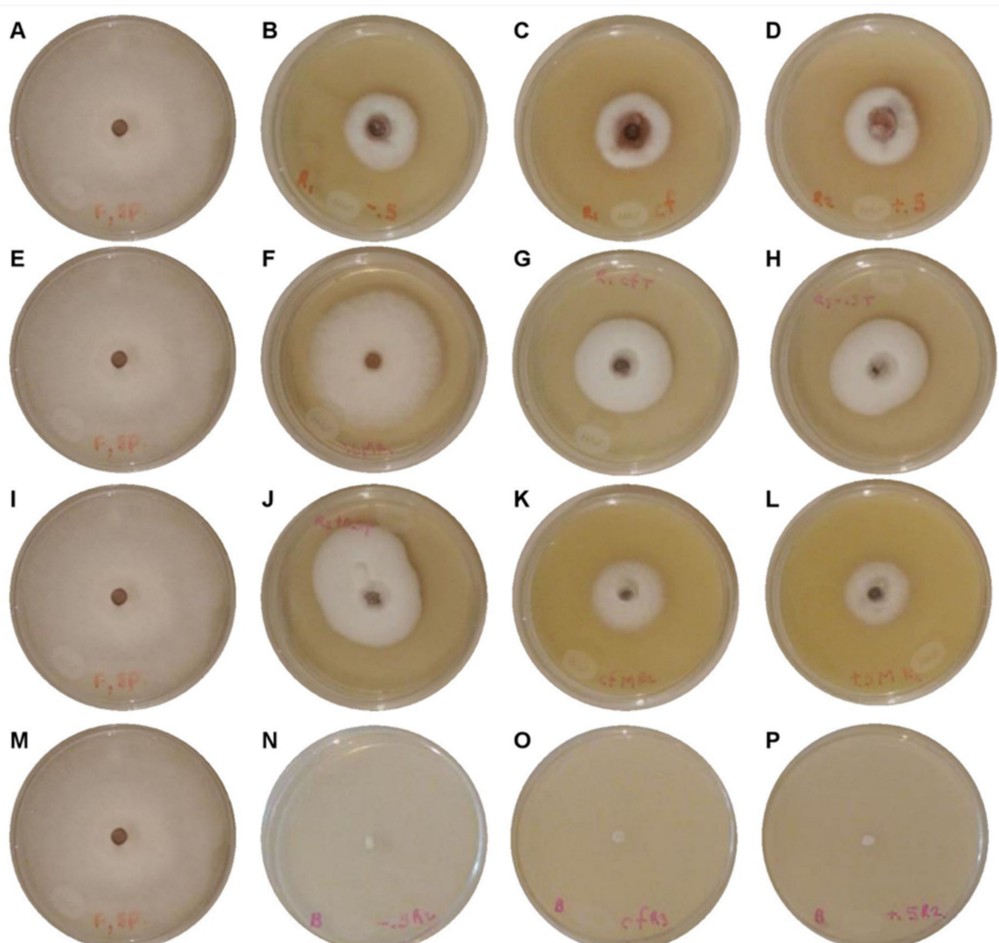

**Figure 4.** Strain MA-FC120 (FSSC) in culture medium with fungicide. (**A**,**E**,**I**,**M**) Controls; Captan 50®: (**B**) 450 mg $L^{-1}$, (**C**) 900 mg $L^{-1}$, (**D**) 1350 mg $L^{-1}$; Mancosol 80®: (**F**) 600 mg $L^{-1}$, (**G**) 1200 mg $L^{-1}$, (**H**) 1800 mg $L^{-1}$; Talonil 75®: (**J**) 450 mg $L^{-1}$, (**K**) 900 mg $L^{-1}$, (**L**) 1350 mg $L^{-1}$; Benomyl 50®: (**N**) 250 mg $L^{-1}$, (**O**) 500 mg $L^{-1}$, and (**P**) 750 mg $L^{-1}$.

At the concentrations evaluated and according to the IOBC [38] classification scale, the protective fungicides Mancosol 80® and Captan 50® were slightly toxic; however, fungicide Talonil 75® at a dose of 1350 mg $L^{-1}$ showed moderate toxicity (Figure 4).

## 4. Discussion

According to the morphological and taxonomic characteristics corresponding to *F. solani* [19,39,40], it was confirmed that the strain MA-FC120 belongs to this fungal genus. Likewise, the phylogenetic analysis of the concatenated sequences (ITS-EF-1α) shows 100% identity with the species *F. solani*. In this regard, Kurt et al. [41] used ITS and EF-1α molecular markers to confirm the identity of CFs4 (ITS: MF972071; EF-1α: MF972074) and CFs8 (ITS: MF972072; EF-1α: MF972075) strains, with both strains belonging to the *F. solani*

species complex (FSSC), associated with citrus dry root rot in the eastern Mediterranean region of Turkey; the same molecular markers were used in the present study.

The phytopathogen *F. solani* can affect more than 100 plant species of agricultural interest, including strawberry crops [42]. It has been reported as a pathogen that causes drying, stunting, wilting, necrosis, and death in strawberry plants [12]. It was detected for the first time in 2012, in strawberry nursery plants with symptoms of crown rot similar to those found in the present study.

Pathogenicity tests revealed similar results to those already reported by Pastrana et al. [12] on strawberry (*Fragaria × ananassa*) plants. Symptoms consisted of wilting of foliage, stunting, drying, and death of plants. Villarino et al. [16] reported vascular wilt, chlorosis, and root rot in strawberry crops. They observed necrotic roots, gray-green wilted leaves, and plant death, symptoms similar to those observed in this study. Fungi of genus *Fusarium* can invade the vascular tissues of plants, impede water transport through the Xylem by inducing vessel blockage and causing wilting of foliage [43]. In addition, *F. solani* concentrates in the collar area, causing rotting and subsequently plant death [44].

The lack of research limits the ability of growers to choose effective products and increases the risk of generating strains resistant to the active ingredients used in fungicides. However, the traditional method of control against pathogenic fungi is still chemical pesticides [45]. Likewise, management of *Fusarium* wilting infection is mainly performed through chemical soil fumigation for strawberry cultivation [46].

Fungicides are commonly used in conventional agriculture to protect strawberries from field rot and improve yield [47]. Da Silva et al. [48] studied in vitro the effect of Mancozeb® on *F. solani* and reported that this pesticide inhibited 42.5% of mycelial growth of pathogens. Another study by Kumar-Gupta et al. [49] reported the effect of Mancozeb® on isolates of *F. solani*, the causal agent of root rot in papaya obtaining 61.46% mycelial-growth inhibition. Both studies presented higher values than those reported in the present research. According to FRAC (2018) [50], Mancozeb® has a multisite mode of action; therefore, it is broad-spectrum and protective, it acts by forming compounds that bind with sulfhydryl groups of amino acids and causes denaturation of proteins and enzymes, and, consequently, suppresses pathogen growth. However, a study by Andrabi et al. [51] reported the effect of Mancozeb® on *F. solani*, the causal agent of root rot in chickpea, where they obtained 54.82% inhibition of mycelial growth, with a concentration of 500 ppm, values similar to those reported in present investigation and considered slightly toxic on the IOBC [38] classification scale. Dithiocarbamate fungicides such as Mancozeb® are among the most widely used pesticides today in the control of a wide variety of diseases in seeds, fruits, and vegetables. However, there are several studies where the fungicide Mancozeb® is related to health diseases such as Parkinson's, teratogenesis, and carcinogenesis, for which reason it is prohibited in several countries [52]. Due to its high toxicity, the minimum purity proposed by the EU NMS is 850 g/kg, while the minimum purity proposed by Agria is 915 g/kg [53].

In the case of the fungicide Captan®, Ayvar-Serna et al. [54] observed that, under in vitro conditions, it inhibited 100% of mycelial growth of *F. solani*, the causal agent of root rot in tomato, at a concentration of 0.044 g. Another study by Shah et al. [55] reported that this pesticide inhibited 75.90% of mycelial growth of pathogens at a concentration of 1000 μg mL$^{-1}$ in vitro. Both studies presented higher values than those reported in the present investigation. Fungicide Captan® presents a fungitoxic compound because it affects energy transport in mitochondria and the physiological functioning of membranes in fungal cells, obstructing the activity of sulfhydryl enzymes and resulting in the release of thiophosgene ($CSCl_2$), a highly toxic compound that kills pathogens [50].

The present study investigated the fungistatic ability of Chlorothalonil®, which is a foliar fungicide with widespread use worldwide [56]. It is a non-systemic organochlorine fungicide. Specifically it is a broad-spectrum polychlorinated aromatic that causes mycelial-growth retardation and inhibits spore germination. It acts on respiration of fungal cells;

that is, it binds to sulfhydryl groups of some amino acids that affect the Krebs cycle by reducing ATP synthesis, causing cell death [57].

Studies by Dwivedi et al. [58] reported that fungicide Chlorothalonil® reduced mycelial growth of *F. solani* by 82.34%, in pathogens isolated from roots in eggplant crops, and these values are similar to those reported in the present investigation, considered moderately toxic on the IOBC [38] rating scale for the highest concentration (1350 mg L$^{-1}$). Bhaliya et al. [59] reported that coriander root rot caused by *F. solani* was controlled by fungicide Chlorothalonil® by 69.96%. A similar finding was observed by Madhusndhan et al. [60] for *F. solani*, the causal agent of root rot in sweet orange, obtaining 62.82% mycelial-growth inhibition, with a concentration of 250 ppm, with values similar to those reported in the present investigation and considered slightly toxic on the IOBC [38] classification scale for the recommended concentration (900 mg L$^{-1}$). For this reason, Salazar et al. [61] report its extensive use in strawberry cultivation under greenhouse conditions in Mexico.

The disappearance of honey bees is only the most visible part of a widespread phenomenon due to massive loss of biodiversity that negatively impacts agroecosystems [62]. The fungicide Chlorothalonil has been shown to alter the intestinal bacterial communities of honey bees, in addition to decreasing larval survival in the hive [63]. However, growers continue to rely on the chemical, without considering the negative effects for the future.

According to the literature, Benomyl® is a systemic fungicide and is banned in developed countries., However, its use is very frequent in developing countries [64], as in the case of Mexico. It is a member of the benzimidazole class, which penetrates the fungal cell and reacts with proteins forming chelates; it also alters intracellular redox balance, inducing oxidative stress, and is a potent inhibitor of DNA synthesis in fungi [65,66].

Recently, Asadboland et al. [67] reported 76% inhibition of mycelial growth of *F. solani* (IRAN 11C) by 0.1 g/mL Benomyl® under in vitro conditions on PDA medium. In this sense, Kordali et al. [68] also reported a reduction in mycelial growth on a strain of *F. solani* from Atatürk University, obtaining 79.8% inhibition with a concentration of 1 mg/mL. However, Dorugade et al. [69] observed that, under in vitro conditions, the fungicide Benomyl® (100 ppm) inhibited 100% mycelial growth of *F. solani* (Fs-3), the causal agent of root rot on elephant-foot sweet potato. These values are like those reported in the present investigation, considered toxic on the IOBC [38] rating scale for the three concentrations used (Figures 4 and 5).

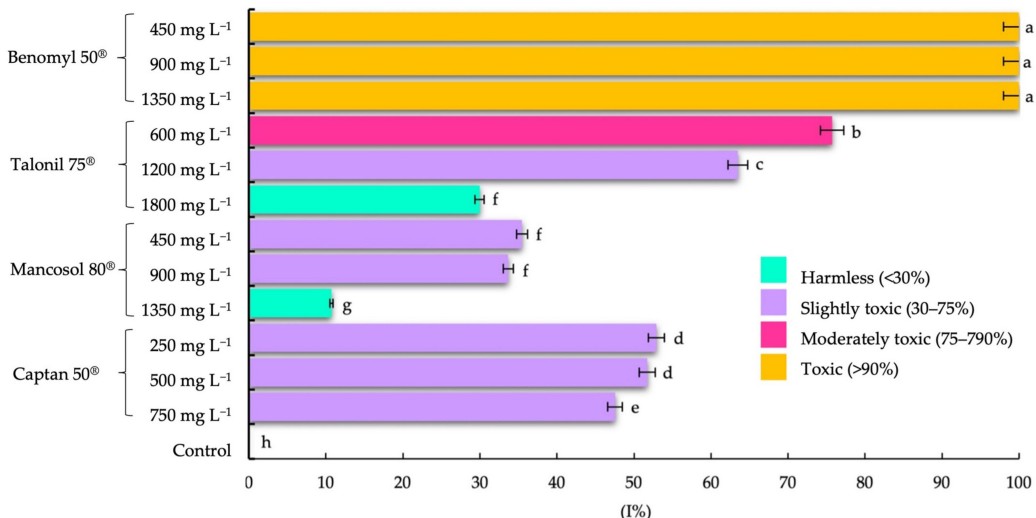

**Figure 5.** The effect of four fungicides at different concentrations according to the classification scale established by IOBC. Means followed by the same letter are not significantly different for $p \leq 0.05$ according to Tukey test. [Reproduced with permission from Viñuela et al., Phytoma; published by PHYTOMA-España, 1993].

Despite advances in international regulation, considerable information gaps and uncertainties remain about the risks of pesticides, particularly for low- and middle-income countries compared to high-income countries [70].

## 5. Conclusions

The MA-FC120 strain belongs to the *Fusarium solani* species complex (FSSC) and is associated with the wilting and root rot of the Camino Real variety of the strawberry crop, located in the rural area of Santa Cruz Analco, belonging to the municipality of San Salvador el Green Puebla-Mexico. The fungicides Mancosol 80® and Talonil 75® at low concentration (600 and 450 mg $L^{-1}$) showed no toxicity, being harmless to strain MA-FC120. However, Talonil 75® showed slight toxicity at the dose recommended by the manufacturer and moderate toxicity at high concentration (1350 mg $L^{-1}$). Likewise, Captan 50® fungicide, in its three concentrations evaluated, showed slight toxicity, obtaining around 50% on the classification scale established by IOBC. Unlike the protective fungicides, systemic fungicide Benomyl 50® showed 100% inhibition of mycelial growth of *F. solani* at the three concentrations evaluated. This research represents the first study of the most widely used fungicides for strawberry production in Mexico and their effect on *F. solani*, obtaining results that can lead to reduction in chemical products and generate less resistance to persistent fungal strains of the strawberry crop variety "Camino Real", from the Santa Cruz Analco community, Puebla-Mexico.

**Author Contributions:** Conceptualization, O.R.-A., C.P.L., and A.C.C.; methodology, A.R.T., A.C.C., and O.R.-A.; software, O.S.T., and O.R.-A.; validation, Y.P.H., A.R.T., and O.R.-A.; formal analysis, A.C.C., C.P.L., and O.R.-A.; resources, O.R.-A. and C.P.L.; original—draft preparation, Y.P.H., O.S.T., and O.R.-A.; writing—review and editing, A.R.T. and O.R.-A.; visualization, O.R.-A. and A.C.C.; supervision, O.S.T.; project administration, O.R.-A.; funding acquisition, O.R.-A. All authors have read and agreed to the published version of the manuscript.

**Funding:** This research was supported by the program PRODEP 2020 of the Secretaría de educación Pública of Mexico (SEP); the Consejo Nacional de Ciencia y Tecnología (CONACyT) number 72760 and Benémerita Universidad Autónoma of Puebla, number 100420500.

**Institutional Review Board Statement:** Not applicable.

**Informed Consent Statement:** Not applicable.

**Data Availability Statement:** Informed consent was obtained from all subjects involved in the study.

**Acknowledgments:** The authors are grateful to Consejo Nacional de Ciencia y Tecnología (CONACyT) and laboratory 204 of the Center for Agroecology at Benémerita Universidad Autónoma of Puebla.

**Conflicts of Interest:** The authors declare no conflict of interest.

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
