# Peer review of "Efficacy of Four In Vitro Fungicides for Control of Wilting of Strawberry Crops in Puebla-Mexico"

_applsci, doi:10.3390/app12073213_

Round 1

Reviewer 1 Report

The item well fits with the Journal purposes having a direct application in agrifood production. The manuscript is well written and conceived but suffers some flaws that should be addressed before considering for publication. The identification of strain MA-FC120 at the species level is not adequately supported by the analysis performed and achieved results. More attention should be paid to the wording (i.e., inhibition/toxicity, see detailed notes below), the title and keywords should be reworked.

Below are some additional notes

The title does not reflect the manuscript content. I mean, the authors started from crop wilt, they isolate and identified the causal agent and then performed in vitro tests. The title instead sounds like someone spoils your book reading by telling the murderer’s name before opening it.

Solani, even if part of the title should not be capitalized

Keywords Please rework this section to improve the paper visibility enhancing their finding during searches. The words in the title are already indexed, so should be avoided. You have up to 10 keywords, don’t waste them. KW are not single words, please take a look here “https://falconediting.com/en/blog/6-tips-for-choosing-keywords-for-your-scientific-manuscript” to have some suggestions on how to.

If you are not certain of the power of your KW, you can googling them on scholar and check if the results fit with your field of expertise. I suggest something like fungicide resistance, plant-pathogen control, broad-spectrum fungicides, and so on

L41-42 Improve readability of numbers “ 3213000 t, followed by USA (1164000 t) and Mexico [4,5]. The cultivated 41 area in Mexico is currently 12913 ha with a production of 557514 “

L44-45 losses…. losses please change to avoid repeats

L90 a cooler until they… please indicate temperature

L95 PDA medium  please indicate the supplier, city, Nation

L96 cultures, following indications of Morales et al.- please describe it in short

L110 for PCR amplification, please change to until further processing or something similar.

L106 HPLC water did you mean ultrapure water (UPW)/ Milli-Q water ?

L111 “MA-FC120” strain. In section 2.2 has been reported the presence of colonies, the same in section 2.3.

Please add a short sentence explaining the ratio that justifies the passage from colonies to a single strain and collection number assignment

Please refer to strain MA-FC120 instead of “MA-FC120” strain.

In the MA-FC120 part of a culture collection, I mean whatever culture collection even personal

L113 and confirmed. Please remove “confirmed” because are necessary both markers and ITS is less important to EF-1alpha for the assignment to Fusarium solani species complex (FSSC).

L115-116 and sent for direct sequencing from both directions to Macrogen, Seoul, Korea. Please change to and bidirectionally sequenced by Macrogen, Seoul, Korea

L120 are these concatenated sequences? Please explain

L120 before the evolutionary model mention you should define the dataset on which you applied that model.

L127 please check consistency writing GenBank, PB should be bp as usually

Table 1 As the authors know, the simple match does not corroborate the identity of strain MA-FC120. I suppose the sequences in Table 1 are those used for the phylogenetic analysis. If so, a number of changes should be done in the table and in the text. How did the authors choose the sequences for analysis? I mean simply match, favouring those from recognized/reliable collections such as ATCC, CBS etc? Are all the FSCC clusters (three) equally represented? What about the outgroup? All information should be reported here.

Figure 2 I can argue that your strain, having an external position, is not a Fusarium solani, or considered the limited selection used it could belong to a different solani cluster. Considering that this is not a phylogenetic study and at this stage, you don’t need to strictly determine his identity, I suggest 1) referring strain MA-FC120 as FSSC. 2) implement the dataset with the closest to FSSC to give an upper and lower limit; 3) please use also reference/type strains

In light of the abovementioned notes, Table 1column headers should be modified as follows

Species name     strain number    isolation source Country                ITS         EF-1alpha. in these last two columns should be used to report the accession numbers.

In this table should be reported also the MA-FC120 strain accession numbers. So, The sequences in  Supplementary material are not necessary

Figure 2 the outgroup should not in bold, the strain in the study should be bold black.

L130 Pathogenicity test. Please give mention to negative control; L138-140 are results and should be removed.

L142-143 what “a.i.” does it mean? Are you indicating the active compound?

Please add the relative CAS numbers

L150 Papa Dextrose Agar, do you mean Potato Dextrose Agar? Please add the supplier

L212-214 the sentence should be shortened avoiding protocol repeats. The figure did not show clearly the identity of strain MA-FC120 as above.

3.3 title should be changed giving a link with materials and methods…using something like In vitro sensivity bioassay and mycelial growth inhibition.

L245 was the best fungicide… I suggest changing it to was the most effective product

Figure 4 add the strain number

Typical growth rate was 6.7104 mm d-1, - Do you mean in Negative Control plates? Please rework

L259 showed a moderately toxic effect on mycelial growth. The author using in vitro test measured the inhibition power of biocide on fungal growth. The authors measured the reduction of colony diameter (visible effect) and not toxicity (cause). So that, reporting results they should refer only to the former and not the latter. Figure 4 should be accordingly amended removing all commenting words

L284-285 reported by Leslie and Summerell [37], Mehmood et al. [19], and Šišić et al. [38]. In this regard, Kurt et al. [39]…. This citation mode did not accomplish the journal instructions. Please check this item along with the text.

Discussion should give more attention to the fact that Mancozeb is a fungicide subject to restrictions. For example, EU banned its use last year. Chlorothalonil has been EU banned in 2019 and so sooner or later the list of products allowed in Mexico could change.

Higher toxicity is related to higher risk for humans and environment and life should have the same weight worldwide. More, it should have mention the importance of pollinators…because without them no fruits comes.

L371-374 The present work is the first study showing the doses and fungicides most used in strawberry cultivation, where their application can lead to reduction of chemical products and generate less resistance to persistent fungal strains, which can jeopardize the effectiveness of the active component.

This sentence is not clear and quite verbose, please rework

Author Response

Thanks for the comments and suggestion.

Reviewer Comments

The item well fits with the Journal purposes having a direct application in agri-food production. The manuscript is well written and conceived but suffers some flaws that should be addressed before considering for publication. The identification of strain MA-FC120 at the species level is not adequately supported by the analysis performed and achieved results. More attention should be paid to the wording (i.e., inhibition/toxicity, see detailed notes below), the title and keywords should be reworked.

I inform you that each of your comments were met to substantially improve the work presented. In the attached file, the changes marked in yellow color are shown.

In relation to the accession No. of the sequence OM616884, it will be released on March 6, 2022. 

Reviewer Comments

  • The title does not reflect the manuscript content. I mean, the authors started from crop wilt, they isolate and identified the causal agent and then performed in vitro tests. The title instead sounds like someone spoils your book reading by telling the murderer’s name before opening it.

Answer

The title is modified at the suggestion of the author, remaining as follows “Efficacy of Four In Vitro Fungicides for Control of Strawberry Crop Wilt in Puebla-Mexico”.

Reviewer Comments

  • Keywords, Please rework this section to improve the paper visibility enhancing their finding during searches. The words in the title are already indexed, so should be avoided. You have up to 10 keywords, don’t waste them. KW are not single words, please take a look here “https://falconediting.com/en/blog/6-tips-for-choosing-keywords-for-your-scientific-manuscript” to have some suggestions on how to.

Answer

The keywords are modified at the author's suggestion, remaining as follows: Fungicide resistance; PCR; broad spectrum fungicides; Fusarium solani; toxicity.

 Reviewer Comments

  • L41-42 Improve readability of numbers “3213000 t, followed by USA (1164000 t) and Mexico [4,5]. The cultivated 41 area in Mexico is currently 12913 ha with a production of 557514 “Answer: Done
  • L44-45 losses…. losses please change to avoid repeats. Answer: Done
  • L90 a cooler until they… please indicate temperature. Answer: Done
  • L95 PDA medium please indicate the supplier, city, Nation. Answer: Done
  • L96 cultures, following indications of Morales et al.- please describe it in short. Answer: Writing is improved
  • L110 for PCR amplification, please change to until further processing or something similar. Answer: Done
  • L106 HPLC water did you mean ultrapure water (UPW)/ Milli-Q water ? Answer: Writing is improved. Yes it is Milli-Q water
  • L111 “MA-FC120” strain. In section 2.2 has been reported the presence of colonies, the same in section 2.3. Please add a short sentence explaining the ratio that justifies the passage from colonies to a single strain and collection number assignment. Answer: The wording of the paragraph is improved and it is explained in more detail.
  • Please refer to strain MA-FC120 instead of “MA-FC120” strain. Answer: Done
  • In the MA-FC120 part of a culture collection, I mean whatever culture collection even personal. Answer: It refers to the collection of pathogenic fungi from the Agroecology Center of the Institute of Sciences of the Benemerita Autonomous University of Puebla, and MA-FC120 refers to its identification number.
  • L113 and confirmed. Please remove “confirmed” because are necessary both markers and ITS is less important to EF-1alpha for the assignment to Fusarium solani species complex (FSSC). Answer: Done
  • L115-116 and sent for direct sequencing from both directions to Macrogen, Seoul, Korea. Please change to and bidirectionally sequenced by Macrogen, Seoul, Korea Answer: Done
  • L120 are these concatenated sequences? Please explain. Answer: Yes, they are concatenated sequences of the two sequenced regions. It is better detailed in the text.
  • L120 before the evolutionary model mention you should define the dataset on which you applied that model. Answer: Done
  • L127 please check consistency writing GenBank, PB should be bp as usually. Answer:
  • Table 1 As the authors know, the simple match does not corroborate the identity of strain MA-FC120. I suppose the sequences in Table 1 are those used for the phylogenetic analysis. If so, a number of changes should be done in the table and in the text. How did the authors choose the sequences for analysis? I mean simply match, favouring those from recognized/reliable collections such as ATCC, CBS etc? Are all the FSCC clusters (three) equally represented? What about the outgroup? All information should be reported here. Answer: Table 1 is modified at the reviewer's suggestion. The accessions were chosen based on the Blast algorithm with sequences established in GenBank™, and that presented the two amplified regions (ITS and TEF-1α) with an identity percentage of 100% and 99.85%, respectively.
  • Figure 2 I can argue that your strain, having an external position, is not a Fusarium solani, or considered the limited selection used it could belong to a different solani cluster. Considering that this is not a phylogenetic study and at this stage, you don’t need to strictly determine his identity, I suggest 1) referring strain MA-FC120 as FSSC. 2) implement the dataset with the closest to FSSC to give an upper and lower limit; 3) please use also reference/type strains. Answer: The concatenated phylogenetic analysis was performed again with different accessions (Table 1) closest to the strain MA-FC120. The analysis revealed 100% identity for F. solani. It is decided to keep the name of the species for this research work.
  • Figure 2 the outgroup should not in bold, the strain in the study should be bold black. Answer: Done
  • L130 Pathogenicity test. Please give mention to negative control; L138-140 are results and should be removed. Answer: Done
  • L142-143 what “a.i.” does it mean? Are you indicating the active compound? Answer: ai." it means? active ingredient. Table 2 is modified with this information.
  • L150 Papa Dextrose Agar, do you mean Potato Dextrose Agar? Please add the supplier. Answer:
  • L212-214 the sentence should be shortened avoiding protocol repeats. Answer: The figure did not show clearly the identity of strain MA-FC120 as above. Figure 2 is modified at the reviewer's suggestion.
  • 3 title should be changed giving a link with materials and methods…using something like In vitro sensivity bioassay and mycelial growth inhibition. Answer: Done.
  • L245 was the best fungicide… I suggest changing it to was the most effective product. Answer:
  • Figure 4 add the strain number. Answer:
  • Typical growth rate was 6.7104 mm d-1, - Do you mean in Negative Control plates? Please rework. Answer: The wording of the statement is improved.
  • L259 showed a moderately toxic effect on mycelial growth. The author using in vitro test measured the inhibition power of biocide on fungal growth. The authors measured the reduction of colony diameter (visible effect) and not toxicity (cause). So that, reporting results they should refer only to the former and not the latter. Figure 4 should be accordingly amended removing all commenting words. Answer: However, in the present investigation the results of the percentage of inhibition are compared with the classification scale established by OILB to determine the effects of the pesticide on the strain MA-FC120 and determine its toxicity. See table 2.
  • L284-285 reported by Leslie and Summerell [37], Mehmood et al. [19], and Šišić et al. [38]. In this regard, Kurt et al. [39]…. This citation mode did not accomplish the journal instructions. Please check this item along with the text. Answer: Writing is improved.
  • Discussion should give more attention to the fact that Mancozeb is a fungicide subject to restrictions. For example, EU banned its use last year. Chlorothalonil has been EU banned in 2019 and so sooner or later the list of products allowed in Mexico could change. Answer: The discussion is improved by adding a paragraph referring to the restrictions of permitted products.
  • Higher toxicity is related to higher risk for humans and environment and life should have the same weight worldwide. More, it should have mentioned the importance of pollinators…because without them no fruits comes. Answer: The discussion is improved by adding a paragraph referring to the importance of pollinators and their effects on them.
  • L371-374 The present work is the first study showing the doses and fungicides most used in strawberry cultivation, where their application can lead to reduction of chemical products and generate less resistance to persistent fungal strains, which can jeopardize the effectiveness of the active component. Answer: Writing is improved

Reviewer 2 Report

This is still premature paper. This paper MUST be completed by doing in vivo work. Doing in vitro work to test the efficiency of these fungicides is NOT enough to publish a paper in international journal. So this paper needs to be continued with in vivo greenhouse and/or field experiments to continue the work. So my decision is to reject the paper because the paper from my point of view is still premature and the data in the paper is NOT enough. Please continue the work, test the effect of these fungicides on the pathogen under in vivo greenhouse and/or field experiments. The work in its current format is incomplete and can NOT be published.

Author Response

Reviewer Comments

This is still premature paper. This paper MUST be completed by doing in vivo work. Doing in vitro work to test the efficiency of these fungicides is NOT enough to publish a paper in international journal. So this paper needs to be continued with in vivo greenhouse and/or field experiments to continue the work. So my decision is to reject the paper because the paper from my point of view is still premature and the data in the paper is NOT enough. Please continue the work, test the effect of these fungicides on the pathogen under in vivo greenhouse and/or field experiments. The work in its current format is incomplete and can NOT be published.

 Answer:

Thank you for your comments and suggestions to improve the document. However, I tell you this: The journal Applied Sciences does consider experimental work with the aim of encouraging scientists to publish their results obtained in the laboratory or field or both. I quote some examples:

https://doi.org/10.3390/app8122577

https://doi.org/10.3390/app11125391

https://doi.org/10.3390/app11125612 

https://doi.org/10.3390/app11125339

In addition, the work is not premature and if it has international relevance, I quote some similar works:

  • Prince Kumar Gupta, S.K Singh, Sneha Shikha. In vitro efficacy of different fungicides against Fusarium solani isolate causing root rot of papaya (Carica papaya L). Int J Chem Stud 2020;8(3):221-224. DOI: 10.22271/chemi.2020.v8.i3c.9229
  • Gabrekiristos E, Ayana G (2020) In Vitro Evaluation of Some Fungicides against Fusarium oxysporum the Causal of Wilt Disease of Hot Pepper (Capsicum annum) in Ethiopia. Adv Crop Sci Tech. 8:443. DOI: 10.35248/2329-8863.20.8.443
  • Mamza, W.S., A.B. Zarafi and O. Alabi. 2010. In vitro evaluation of six fungicides on radial mycelial growth and regrowth of Fusarium pallidoroseum isolated from castor (Ricinus communis) in Samaru, Nigeria. Archives Phytopathol. Plant Protect., 43(2):116-122.
  • Bashir, M.R., Atiq, M., Sajid, M. et al. Antifungal exploitation of fungicides against Fusarium oxysporum sp. capsici causing Fusarium wilt of chili in Pakistan. Environ Sci Pollut Res 25, 6797–6801 (2018). https://doi.org/10.1007/s11356-017-1032-9
  • Yadav, S. and M.M. Ansari (2017). Isolation, Identification and In vitro Evaluation of Fungicides against Fusarium Leaf Blight of Soybean Caused by oxysporum. Soybean Research, 46.
  • Yaqub F, Shahzad S. Effect of fungicides on in vitro growth of Sclerotium rolfsii. Pak J. Bot, 2006; 38(38): 881-883
  • Chaurasia, A. K., Chaurasia, S., Chaurasia, S., and Chaurasia, S. (2014). In vitro efficacy of fungicides against the growth of foot-rot pathogen (Sclerotium rolfsii) of Brinjal. Int. J. Curr. microbiol. App. Sci, 3(12), 477-485.

I would appreciate your reconsideration to review the work.

Regards

Reviewer 3 Report

Coronel et al. in their manuscript titled “Efficacy of four in vitro fungicides for control of Fusarium solani:causal agent of strawberry crop wilt in Puebla-Mexico” fulfill Koch’s postulates to confirm the role of a F. solani isolate in causing the observed disease in strawberries grown in Mexico.  The authors used genotyping and a small phylogenetic analysis with Genbank accessions of F. solani to demonstrate that the isolate causing disease is appropriately taxonomically classified as F. solani.  The authors then evaluated the response of the F. solani isolate in response to several fungicides in controlled in-vitro studies.  The authors examined isolate growth in PDA amended plates, tracking the rate of colony growth in comparison to the same isolate grown in the absence of fungicide amended plates. The fungicides were mostly applied according to the label dose.  They found that Benomyl had, by far, the greatest impact on reducing in vitro F. solani isolate growth. I appreciate that the authors pointed out that Benomyl is banned in other countries.

The manuscript was well written, and the authors communicated their ideas, methods, rationale, and results clearly. 

I think that there are areas of weakness, that if addressed by the authors will result in a more robust study. 

1A). The use of a single isolate of F. solani for the entire study may or may not be representative of the expected results for strawberry disease control.  My concern lies within the taxon F. solani.  Fusarium solani is a complex taxon with isolates that vary significantly in their ability to cause disease, ranging on a spectrum of host-pathogen interactions from entirely non-pathogenic to obvious disease causal agents.  Because the authors focus on of what appeared to be 16 F. solani isolates collected from field samples, it is possible that the other isolates may not cause disease or respond to fungicides in the same manner.

On lines 205-206, the authors state that they selected the single study isolate because it was “most representative” of their multiple isolates.  I think it is necessary to describe the criteria used to determine why this isolate was most representative, as this choice seems crucial in the authors’ argument to infer fungicide responses across all F. solani isolates.

1B). The reliance on a single isolate for inferring the causal agent of disease in strawberries could be addressed by the authors directly if they have genotyped the other F. solani isolates from their field collection.  It would seem to me that the authors might be able to address whether their single isolate is the dominant disease-causing isolate by comparing the sequences of their study isolate to the other field isolates.  The authors already have a phylogenetic framework to evaluate the other isolates.  I think that this is an important aspect of the study that if addressed would make a more robust study.  However, if the standing genetic variation in F. solani suggests the simultaneous presence of multiple different genetic lineages, the fungicide response assays based on a single isolate may or may not be generalizable.

2). I am not sure if the authors can address this issue or not.  But, in their methodologies for quantifying the response of the F. solani isolate to fungicides on PDA amended plates, they measure colony diameter to populate their quadratic model of growth.  However, in their figures of colony responses, I noted that many of the colonies were oval (not round).  At least, the authors should explain how they selected the colony diameter (average in multiple directions, the smallest diameter, the largest diameter).  Perhaps I missed this description though.  I don’t believe the ultimate conclusion that Benomyl was the most effective fungicide at reducing the F. solani isolate growth (these results are obvious), is impacted by potentially imprecise measures of colony diameter.  However, when comparing the other 3 preventative fungicides to each other, the precision of colony measurement may be obscuring some of the response patterns.  

Author Response

Reply to reviewer 

Coronel et al. in their manuscript titled “Efficacy of four in vitro fungicides for control of Fusarium solani: causal agent of strawberry crop wilt in Puebla-Mexico” fulfill Koch’s postulates to confirm the role of a F. solani isolate in causing the observed disease in strawberries grown in Mexico.  The authors used genotyping and a small phylogenetic analysis with Genbank accessions of F. solani to demonstrate that the isolate causing disease is appropriately taxonomically classified as F. solani.  The authors then evaluated the response of the F. solani isolate in response to several fungicides in controlled in-vitro studies.  The authors examined isolate growth in PDA amended plates, tracking the rate of colony growth in comparison to the same isolate grown in the absence of fungicide amended plates. The fungicides were mostly applied according to the label dose.  They found that Benomyl had, by far, the greatest impact on reducing in vitro F. solani isolate growth. I appreciate that the authors pointed out that Benomyl is banned in other countries.

Thank you for the feedback on the research paper.

I think that there are areas of weakness, that if addressed by the authors will result in a more robust study.

The manuscript fits very well with the scope of the special issue of the journal (Engineering of Smart Agriculture). Likewise, I inform you that each of his comments were addressed to substantially improve the work presented. The attached file shows the changes marked in green.

 Reviewer Comments

1A). The use of a single isolate of F. solani for the entire study may or may not be representative of the expected results for strawberry disease control.  My concern lies within the taxon F. solani.  Fusarium solani is a complex taxon with isolates that vary significantly in their ability to cause disease, ranging on a spectrum of host-pathogen interactions from entirely non-pathogenic to obvious disease causal agents.  Because the authors focus on of what appeared to be 16 F. solani isolates collected from field samples, it is possible that the other isolates may not cause disease or respond to fungicides in the same manner.

Answer:

His observation is true; however, the work is part of a larger investigation. The final work reported not only F. solani, but also Fusarium oxysporum (OM473290), Botrytis cinerea (OM473288), Pestalotiopsis sp., (OM473289), Arcopilus aureus (MW318995), and Neurospora dictyophora (MW520112). Of this small diversity, Arcopilus aureus (MW318995) does not cause disease, however we found that it is a new source of resveratrol production (https://doi.org/10.3390/app11104583). Neurospora dictyophora (MW520112) is under evaluation (PDIS-01-22-0189-PDN) in the journal Plant Disease; since it is the cause of dry rot in strawberry fruits in Mexico and has not been reported yet. You can check the accession Nos. in the Ganbank database.

I can guarantee that the 10 isolates morphologically characterized as F. solani are representative of each other and pathogenic. However, due to economic reasons and Covid-19 pandemic, only the most representative strain of the 10 isolates was selected (This selection is detailed in the document). The sequencing of the amplification of two regions and the phylogenetic analysis of these concatenated regions, showed 100% similarity with F. solani (figure 2) with other species of F. solani and in various countries (Table 1).

 Reviewer Comments

On lines 205-206, the authors state that they selected the single study isolate because it was “most representative” of their multiple isolates.  I think it is necessary to describe the criteria used to determine why this isolate was most representative, as this choice seems crucial in the authors’ argument to infer fungicide responses across all F. solani isolates.

Answer:

The criteria used to determine why the MA-FC120 strain was chosen and named as the most representative in the document are described.

In methodology, the criteria were:

"Once the isolates were characterized, the most representative was selected based on its ability to grow in the three growth media tested, a) potato dextrose agar medium (PDA, Bioxon, Becton Dickinson and Company, Querétaro, México), b) tryptone soy agar (TSA, Sigma-Aldrich, Mexico [Báez-Vallejo, 2020]) and c) water agar added with carnation 8 g/L (KWA [Fisher, 1983]).All culture media were incubated with a photoperiod of 12:12 L:D (12 h of light: 12 h of darkness) for 10 days [2], likewise, the diameter of the mycelium was measured every 12 h with a digital vernier (CD-6 Mitutoyo) to estimate the rate of growth (mm d-1) [32]."

In results were:

"The MA-FC120 isolate was chosen based on its high rate of development and growth speed in three culture media tested (PDA, TSA and MEA). In addition to presenting the highest average of morphological characterization."

 Reviewer Comments

2). I am not sure if the authors can address this issue or not.  But, in their methodologies for quantifying the response of the F. solani isolate to fungicides on PDA amended plates, they measure colony diameter to populate their quadratic model of growth.  However, in their figures of colony responses, I noted that many of the colonies were oval (not round).  At least, the authors should explain how they selected the colony diameter (average in multiple directions, the smallest diameter, the largest diameter).

Answer:

In section 2.7. the missing information is completed and the estimation of the growth rate is better detailed. The following text marked in green is attached:

"The diameter of the mycelium was measured every 12 h with a digital vernier (CD-6 Mitutoyo) to estimate the growth rate (mm d-1), which was calculated with the linear growth function [32] Equation (1), obtaining the average of four measurements of the longitudinal diameter per experimental unit".

Reviewer Comments

Perhaps I missed this description though.  I don’t believe the ultimate conclusion that Benomyl was the most effective fungicide at reducing the F. solani isolate growth (these results are obvious), is impacted by potentially imprecise measures of colony diameter.  However, when comparing the other 3 preventative fungicides to each other, the precision of colony measurement may be obscuring some of the response patterns. 

Answer:

 In the conclusions, the wording of the text is improved to avoid obvious results. It is changed to:

"The systemic fungicide Benomyl 50® showed 100% inhibition of the mycelial growth of F. solani in the three concentrations evaluated".

Round 2

Reviewer 1 Report

The authors accomplished almost all points evidenced last time.

Unfortunately, the figure with phylogeny is broken within two pages. But considering the table, no strains to support the identity of the strain studied as a member of the Fusarium solani 'sensu stricto' group has been added and neither has been specified in the text the membership to Fusarium solani complex group (last time I suggested this solution).

Only this point is left to ament before publication

Author Response

I inform you that each of your comments were taken care of.

Regarding the identity of the strain studied, the sequences ITS: OM473287 and EF-1 α: OM616884 were deposited in the gene bank, confirming that it is F. solani belonging to the Fusarium species complex. In addition, reference is made in the document to the relevance of the MA-FC120 strain to the Fusarium species complex (FSSC).

The attached file shows the changes marked in yellow.

Reviewer 2 Report

The authors improved the paper significantly and it can now be accepted for publication. Although I rejected the paper in round 1. Now This paper can be accepted for publication as the authors improved the paper significantly 

Author Response

I appreciate your consideration to review the work.

As you mention, the work has improved significantly, thanks for your comments

Greetings

Reviewer 3 Report

I think the authors made edits to the revised manuscript that improved its quality from the original submission.

Author Response

(The authors gave the same response as above.)
